# Guided Meta-Policy Search

**Russell Mendonca, Abhishek Gupta, Rosen Kralev, Pieter Abbeel, Sergey Levine, Chelsea Finn**
Department of Electrical Engineering and Computer Science
University of California, Berkeley
{russellm, cbfinn}@berkeley.edu
{abhigupta, pabbeel, svlevine}@eecs.berkeley.edu
rdkralev@gmail.com

## Abstract

Reinforcement learning (RL) algorithms have demonstrated promising results on complex tasks, yet often require impractical numbers of samples since they learn from scratch. Meta-RL aims to address this challenge by leveraging experience from previous tasks so as to more quickly solve new tasks. However, in practice, these algorithms generally also require large amounts of on-policy experience during the *meta-training* process, making them impractical for use in many problems. To this end, we propose to learn a reinforcement learning procedure in a federated way, where individual off-policy learners can solve the individual meta-training tasks, and then consolidate these solutions into a single meta-learner. Since the central meta-learner learns by imitating the solutions to the individual tasks, it can accommodate either the standard meta-RL problem setting, or a hybrid setting where some or all tasks are provided with example demonstrations. The former results in an approach that can leverage policies learned for previous tasks without significant amounts of on-policy data during meta-training, whereas the latter is particularly useful in cases where demonstrations are easy for a person to provide. Across a number of continuous control meta-RL problems, we demonstrate significant improvements in meta-RL sample efficiency in comparison to prior work as well as the ability to scale to domains with visual observations.

## 1 Introduction

Meta-learning is a promising approach for using previous experience across a breadth of tasks to significantly accelerate learning of new tasks. Meta-reinforcement learning considers this problem specifically in the context of learning new behaviors through trial and error with only a few interactions with the environment by building on previous experience. Building effective meta-RL algorithms is critical towards building agents that are *flexible*, such as an agent being able to manipulate new objects in new ways without learning from scratch for each new object and goal. Being able to reuse prior experience in such a way is arguably a fundamental aspect of intelligence.

Enabling agents to adapt via meta-RL is particularly useful for acquiring behaviors in real-world situations with diverse and dynamic environments. However, despite recent advances [7, 8, 17], current meta-RL methods are often limited to simpler domains, such as relatively low-dimensional continuous control tasks [8, 44] and navigation with discrete action commands [7, 24]. Optimization stability and sample complexity are major challenges for the meta-training phase of these methods, with some recent techniques requiring upto 250 million transitions for meta-learning in tabular MDPs [7], which typically require a fraction of a second to solve in isolation.

We make the following observation in this work: while the goal of meta-reinforcement learning is to acquire fast and efficient reinforcement learning procedures, those procedures themselves do not need to be acquired through reinforcement learning directly. Instead, we can use a significantly more stable and efficient algorithm for providing supervision at the meta-level. In this work we show that a practical choice is to use supervised imitation learning. A meta-reinforcement learning algorithm

can receive more direct supervision during *meta-training*, in the form of expert actions, while still optimizing for the ability to quickly learn tasks via reinforcement. Crucially, these expert policies can themselves be produced automatically by standard reinforcement learning methods, such that no additional assumptions on supervision are actually needed. They can also be acquired using very efficient off-policy reinforcement learning algorithms which are otherwise challenging to use with meta-reinforcement learning. When available, incorporating human-provided demonstrations can enable even more efficient meta-training, particularly in domains where demonstrations are easy to collect. At meta-test time, when faced with a new task, the method solves the same problem as conventional meta-reinforcement learning: acquiring the new skill using only reward signals.

Our main contribution is a meta-RL method that learns fast reinforcement learning via supervised imitation. We optimize for a set of parameters such that only one or a few gradient steps leads to a policy that matches the expert's actions. Since supervised imitation is stable and efficient, our approach can gracefully scale to visual control domains and high-dimensional convolutional networks. By using demonstrations during meta-training, there is less of a challenge with exploration in the meta-optimization, making it possible to effectively learn how to learn in sparse reward environments. While the combination of imitation and RL has been explored before [30, 20], the combination of imitation and RL in a *meta-learning* context has not been considered previously. As we show in our experiments, this combination is in fact extremely powerful: compared to meta-RL, our method can meta-learn comparable adaptation skills with up to 10x fewer interaction episodes, making meta-RL much more viable for real-world learning. Our experiments also show that, through our method, we can adapt convolutional neural network policies to new goals through trial-and-error, with only a few gradient descent steps, and adapt policies to sparse-reward manipulation tasks with a handful of trials. We believe this is a significant step towards making meta-RL practical for use in complex real-world environments.

## 2  Related Work

Our work builds upon prior work on meta-learning [39, 1, 47], where the goal is to learn how to learn efficiently. We focus on the particular case of meta-reinforcement learning [39, 7, 48, 8, 24, 14]. Prior methods learned reinforcement learners represented by a recurrent or recursive neural network [7, 48, 24, 41, 33], using gradient descent from a learned initialization [8, 14, 36], using a learned critic that provides gradients to the policy [44, 17], or using a planner and an adaptable model [5, 38]. In contrast, our approach aims to leverage supervised learning for meta-optimization rather than relying on high-variance algorithms such as policy gradient. We decouple the problem of obtaining expert trajectories for each task from the problem of learning a fast RL procedure. This allows us to obtain expert trajectories using efficient, off-policy RL algorithms. Recent work has used amortized probabilistic inference [34] to achieve data-efficient meta-training, however such contextual methods cannot continually adapt to out of distribution test tasks. Further, the ability of our method to utilize example demonstrations if available enables much better performance on challenging sparse reward tasks. Our approach is also related to few-shot imitation [6, 11], in that we leverage supervised learning for meta-optimization. However, in contrast to these methods, we lean an automatic reinforcement learner, which can learn using only rewards and does not require demonstrations for new tasks.

Our algorithm performs meta-learning by first individually solving the tasks with local learners, and then consolidating them into a central meta-learner. This resembles methods like guided policy search, which also use local learners [37, 29, 46, 28, 12]. However, while these prior methods aim to learn a single policy that can solve all of the tasks, our approach instead aims to meta-learn a single learner that can *adapt* to the training task distribution, and generalize to adapt to new tasks that were not seen during training.

Prior methods have also sought to use demonstrations to make standard reinforcement learning more efficient in the single-task setting [30, 20, 21, 45, 4, 42, 16, 43, 32, 26, 19, 40]. These methods aim to learn a policy from demonstrations and rewards, using demonstrations to make the RL problem easier. Our approach instead aims to leverage demonstrations to learn how to efficiently reinforcement learn new tasks without demonstrations, learning new tasks only through trial-and-error. The version of our algorithm where data is aggregated across iterations, is an extension of the DAgger algorithm [35] into the meta-learning setting, and this allows us to provide theoretical guarantees on performance.

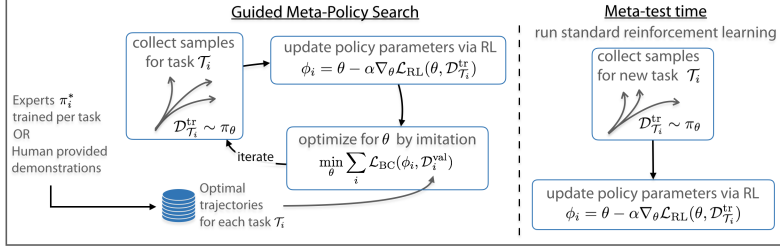

Figure 1: Overview of the guided meta-policy search algorithm: We learn a policy $\pi_\theta$ which is capable of fast adaptation to new tasks via reinforcement learning, by using reinforcement learning in the inner loop of optimization and supervised learning in the meta-optimization. This algorithm either trains per-task experts $\pi_i^*$ or assumes that they are provided by human demonstrations, and then uses this for meta-optimization. Importantly, when faced with a new task we can simply perform standard reinforcement learning via policy gradient, and the policy will quickly adapt to new tasks because of the meta-training.

## 3 Preliminaries

In this section, we introduce the meta-RL problem and overview model-agnostic meta-learning (MAML) [8], which we build on in our work. We assume a distribution of tasks $\mathcal{T} \sim p$, where meta-training tasks are drawn from $p$ and meta-testing consists of learning held-out tasks sampled from $p$ through trial-and-error, by leveraging what was learned during meta-training. Formally, each task $\mathcal{T} = \{r(\mathbf{s}_t, \mathbf{a}_t), q(\mathbf{s}_1), q(\mathbf{s}_{t+1}|\mathbf{s}_t, \mathbf{a}_t)\}$ consists of a reward function $r(\mathbf{s}_t, \mathbf{a}_t) \to \mathbb{R}$, an initial state distribution $q(\mathbf{s}_1)$, and unknown dynamics $q(\mathbf{s}_{t+1}|\mathbf{s}_t, \mathbf{a}_t)$. The state space, action space, and horizon $H$ are shared across tasks. Meta-learning methods learn using experience from the meta-training tasks, and are evaluated on their ability to learn new meta-test tasks. MAML in particular performs meta-learning by optimizing for a deep network's initial parameter setting such that one or a few steps of gradient descent on a small dataset leads to good generalization. Then, after meta-training, the learned parameters are fine-tuned on data from a new task.

Concretely, consider a supervised learning problem with a loss function denoted as $\mathcal{L}(\theta, \mathcal{D})$, where $\theta$ denotes the model parameters and $\mathcal{D}$ denotes the labeled data. During meta-training, a task $\mathcal{T}$ is sampled, along with data from that task, which is randomly partitioned into two sets, $\mathcal{D}^{\text{tr}}$ and $\mathcal{D}^{\text{val}}$. MAML optimizes for a set of model parameters $\theta$ such that one or a few gradient steps on $\mathcal{D}^{\text{tr}}$ produces good performance on $\mathcal{D}^{\text{val}}$. Thus, using $\phi_\mathcal{T}$ to denote the updated parameters, the MAML objective is the following:

$$\min_\theta \sum_\mathcal{T} \mathcal{L}(\theta - \alpha \nabla_\theta \mathcal{L}(\theta, \mathcal{D}_\mathcal{T}^{\text{tr}}), \mathcal{D}_\mathcal{T}^{\text{val}}) = \min_\theta \sum_\mathcal{T} \mathcal{L}(\phi_\mathcal{T}, \mathcal{D}_\mathcal{T}^{\text{val}}).$$

where $\alpha$ is a step size that can be set as a hyperparameter or learned. Moving forward, we will refer to the outer objective as the *meta-objective*. Subsequently, at meta-test time, $K$ examples from a new, held-out task $\mathcal{T}_{\text{test}}$ are presented and we can run gradient descent starting from $\theta$ to infer model parameters for the new task: $\phi_{\mathcal{T}_{\text{test}}} = \theta - \alpha \nabla_\theta \mathcal{L}(\theta, \mathcal{D}_{\mathcal{T}_{\text{test}}}^{\text{tr}})$.

The MAML algorithm can also be applied to the meta-reinforcement learning setting, where each dataset $\mathcal{D}_{\mathcal{T}_i}$ consists of trajectories of the form $\mathbf{s}_1, \mathbf{a}_1, ..., \mathbf{a}_{H-1}, \mathbf{s}_H$ and where the inner and outer loss function corresponds to the negative expected reward:

$$\mathcal{L}_{\text{RL}}(\phi, \mathcal{D}_{\mathcal{T}_i}) = -\frac{1}{|\mathcal{D}_{\mathcal{T}_i}|} \sum_{\mathbf{s}_t, \mathbf{a}_t \in \mathcal{D}_{\mathcal{T}_i}} r_i(\mathbf{s}_t, \mathbf{a}_t) = -\mathbb{E}_{\mathbf{s}_t, \mathbf{a}_t \sim \pi_\phi, q_{\mathcal{T}_i}} \left[ \frac{1}{H} \sum_{t=1}^H r_i(\mathbf{s}_t, \mathbf{a}_t) \right]. \quad (1)$$

Policy gradients [49] are used to estimate the gradient of this loss function. Thus, the algorithm proceeds as follows: for each task $\mathcal{T}_i$, first collect samples $\mathcal{D}_{\mathcal{T}_i}^{\text{tr}}$ from the policy $\pi_\theta$, then compute the updated parameters using the policy gradient evaluated on $\mathcal{D}_{\mathcal{T}_i}^{\text{tr}}$, then collect new samples $\mathcal{D}_{\mathcal{T}_i}^{\text{val}}$ via the updated policy parameters, and finally update the initial parameters $\theta$ by taking a gradient step on the meta-objective. In the next section, we will introduce a new approach to meta-RL that incorporates a more stable meta-optimization procedure that still converges to the same solution under some regularity assumptions, and that can naturally leverage demonstrations or policies learned for previous tasks if desired.

## 4 Guided Meta-Policy Search

Existing meta-RL algorithms generally perform meta-learning from scratch with on-policy methods. This typically requires a large number of samples during meta-training. What if we instead formulate

meta-training as a data-driven process, where the agent had previously learned a variety of tasks with standard multi-task reinforcement learning techniques, and now must use the data collected from those tasks for meta-training? Can we use this experience or these policies in meaningful ways during meta-training? Our goal is to develop an approach that can use these previously learned skills to guide the meta-learning process. While we will still require on-policy data for inner loop sampling, we will require considerably less of it than what we would need without using this prior experience. Surprisingly, as we will show in our experiments, separating meta-training into two phases in this way – a phase that individually solves the meta-training tasks and a second phase that uses them for meta-learning – actually requires less total experience overall, as the individual tasks can be solved using highly-efficient off-policy reinforcement learning methods that actually require less experience taken together than a single meta-RL training phase. We can also improve sample efficiency during meta-training even further by incorporating explicit demonstrations. In the rest of this section, we describe our approach, analyze its theoretical properties, and discuss its practical implementation in multiple real world scenarios.

## 4.1 Guided Meta-Policy Search Algorithm

In the first phase of the algorithm, task learning, we learn policies for each of the meta-training tasks. While these policies solve the meta-training tasks, they do not accelerate learning of future meta-test tasks. In Section 4.3, we describe how these policies are trained. Instead of learning policies explicitly through reinforcement learning, we can also obtain expert demonstrations from a human demonstrator, which can be used equivalently with the same algorithm. In the second phase, meta-learning, we will learn to reinforcement learn using these policies as supervision at the meta-level. In particular, we train for a set of initial parameters $\theta$ such that only one or a few steps of gradient descent produces a policy that matches the policies learned in the first phase.

We will denote the optimal or near-optimal policies learned during the task-learning phase for each meta-training task $\mathcal{T}_i$ as $\{\pi_i^*\}$. We will refer to these individual policies as "experts," because after the first phase, they represent optimal or near-optimal solutions to each of the tasks. Our goal in the meta-learning phase is to optimize the same meta-objective as the MAML algorithm, $\mathcal{L}_{\text{RL}}(\phi_i, \mathcal{D}_i)$, where $\phi_i$ denotes the parameters of the policy adapted to task $\mathcal{T}_i$ via gradient descent. The inner policy optimization will remain the same as the policy-gradient MAML algorithm; however, we will optimize this meta-objective by leveraging the policies learned in the first phase. In particular, we will base the outer objective on supervised imitation, or behavior cloning (BC), of expert actions. The behavioral cloning loss function we use is $\mathcal{L}_{\text{BC}}(\phi_i, \mathcal{D}_i) \triangleq -\sum_{(\mathbf{s}_t, \mathbf{a}_t) \in \mathcal{D}} \log \pi_\phi(\mathbf{a}_t \mid \mathbf{s}_t)$.

Gradients from supervised learning are lower variance, and hence more stable than reinforcement learning gradients [27]. The specific implementation of the second phase proceeds as follows: we first roll out each of the policies $\pi_i^*$ to collect a dataset of expert trajectories $\mathcal{D}_i^*$ for each of the meta-training tasks $\mathcal{T}_i$. Using this initial dataset, we update our policy according to the following meta-objective:

$$\min_\theta \sum_{\mathcal{T}_i} \sum_{\mathcal{D}_i^{\text{val}} \sim \mathcal{D}_i^*} \mathbb{E}_{\mathcal{D}_i^{\text{tr}} \sim \pi_\theta} \left[ \mathcal{L}_{\text{BC}}(\theta - \alpha \nabla_\theta \mathcal{L}_{\text{RL}}(\theta, \mathcal{D}_i^{\text{tr}}), \mathcal{D}_i^{\text{val}}) \right]. \qquad (2)$$

We discuss how this objective can be efficiently optimized in Section 4.3. The result of this optimization is a set of initial policy parameters $\theta$ that can adapt to a variety of tasks, to produce $\phi_i$, in a way that comes close to the expert policy's actions. Note that, so far, we have not actually required querying the expert beyond access to the initial rollouts; hence, this first step of our method is applicable to problem domains where demonstrations are available in place of learned expert policies. However, when we do have policies for the meta-training tasks, we can continue to improve. In particular, while supervised learning provides stable, low-variance gradients, behavior cloning objectives are prone to compounding errors. In the single task imitation learning setting, this issue can be addressed by collecting additional data from the learned policy, and then labeling the visited states with optimal actions from the expert policy, as in DAgger [35]. We can extend this idea to the meta-learning setting by alternating between data aggregation into dataset $\mathcal{D}^*$ and meta-policy optimization in Eq. 2. Data aggregation entails (1) adapting the current policy parameters $\theta$ to each of the meta-training tasks to produce $\{\phi_i\}$, (2) rolling out the current adapted policies $\{\pi_{\phi_i}\}$ to produce states $\{\{\mathbf{s}_t\}_i\}$ for each task, (3) querying the experts to produce supervised data $\mathcal{D} = \{\{(\mathbf{s}_t, \pi_i^*(\mathbf{s}_t)\}_i\}$, and finally (4) aggregating this data with the existing supervised data $\mathcal{D}^* \leftarrow \mathcal{D}^* \bigcup \mathcal{D}$. This meta-training algorithm is summarized in Alg. 1, and analyzed in Section 4.2. When provided with new tasks at meta-test time, we initialize $\pi_\theta$ and run the policy gradient algorithm.

| **Algorithm 1** GMPS: Guided Meta-Policy Search | **Algorithm 2** Optimization of Meta Objective |
|---|---|
| **Require:** Set of meta-training tasks $\{\mathcal{T}_i\}$ | **Require:** Set of meta-training tasks $\{\mathcal{T}_i\}$ |
| 1: Use RL to acquire $\pi_i^*$ for each meta-training task $\mathcal{T}_i$ | **Require:** Aggregated dataset $\mathcal{D}^* := \{\mathcal{D}_i^*\}$ |
| 2: Initialize $\mathcal{D}^* = \{\mathcal{D}_i^*\}$ with roll-outs from each $\pi_i^*$ | **Require:** $\theta$ initial parameters |
| 3: Randomly initialize $\theta$ | 1: **while** not done **do** |
| 4: **while** not done **do** | 2:    Sample task $\mathcal{T}_i \sim \{\mathcal{T}_i\}$ {or minibatch of tasks} |
| 5:    Optimize meta-objective in Eq. 2 w.r.t. $\theta$ using Alg. 2 with aggregated data $\mathcal{D}^*$ | 3:    Sample $K$ roll-outs $\mathcal{D}_i^{\text{tr}} = \{(\mathbf{s}_1, \mathbf{a}_1, ... \mathbf{s}_H)\}$ with $\pi_\theta$ in $\mathcal{T}_i$ |
| 6:    **for** each meta-training task $\mathcal{T}_i$ **do** | 4:    $\theta_{\text{init}} \leftarrow \theta$ |
| 7:       Collect $\mathcal{D}_i^{\text{tr}}$ as $K$ roll-outs from $\pi_\theta$ in task $\mathcal{T}_i$ | 5:    **for** $n = 1...N_{\text{BC}}$ **do** |
| 8:       Compute task-adapted parameters with gradient descent: $\phi_i = \theta - \alpha \nabla_\theta \mathcal{L}_{\text{RL}}(\theta, \mathcal{D}_i^{\text{tr}})$ | 6:       Evaluate $\nabla_\theta \mathcal{L}_{\text{RL}}(\theta, \mathcal{D}_i^{\text{tr}})$ according to Eq. 3 with importance weights $\frac{\pi_\theta(\mathbf{a}_t \mid \mathbf{s}_t)}{\pi_{\theta_{\text{init}}}(\mathbf{a}_t \mid \mathbf{s}_t)}$ |
| 9:       Collect roll-outs from $\pi_{\phi_i}$, resulting in data $\{(\mathbf{s}_t, \mathbf{a}_t)\}$ | 7:       Compute adapted parameters with gradient descent: $\phi_i = \theta - \alpha \nabla_\theta \mathcal{L}_{\text{RL}}(\theta, \mathcal{D}_i^{\text{tr}})$ |
| 10:      Aggregate $\mathcal{D}_i^* \leftarrow \mathcal{D}_i^* \bigcup \{(\mathbf{s}_t, \pi_i^*(\mathbf{s}_t))\}$ | 8:       Sample expert trajectories $\mathcal{D}_i^{\text{val}} \sim \mathcal{D}_i^*$ |
| 11:    **end for** | 9:       Update $\theta \leftarrow \theta - \beta \nabla_\theta \mathcal{L}_{\text{BC}}(\phi_i, \mathcal{D}_i^{\text{val}})$. |
| 12: **end while** | 10:   **end for** |
| | 11: **end while** |

Our algorithm, which we call guided meta-policy search (GMPS), has appealing properties that arise from decomposing the meta-learning problem explicitly into the task learning phase and the meta-learning phase. This decomposition enables the use of previously learned policies or human-provided demonstrations. We find that it also leads to increased stability of training. Lastly, the decomposition makes it easy to leverage privileged information that may only be available during meta-training such as shaped rewards, task information, low-level state information such as the positions of objects [23]. In particular, this privileged information can be provided to the initial policies as they are being learned and hidden from the meta-policy such that the meta-policy can be applied in test settings where such information is not available. This technique makes it straightforward to learn vision-based policies, for example, as the bulk of learning can be done without vision, while visual features are learned with supervised learning in the second phase. Our method also inherits appealing properties from policy gradient MAML, such as the ability to continue to learn as more and more experience is collected, in contrast to recurrent neural networks that cannot be easily fine-tuned on new tasks.

## 4.2 Convergence Analysis

Now that we have derived a meta-RL algorithm that leverages supervised learning for increased stability, a natural question is: will the proposed algorithm converge to the same answer as the original (less stable) MAML algorithm? Here, we prove that GMPS with data aggregation, described above, will indeed obtain near-optimal cumulative reward when supplied with near-optimal experts. Our proof follows a similar technique to prior work that analyzes the convergence of imitation algorithms with aggregation [35, 18], but extends these results into the meta-learning setting. More specifically, we can prove the following theorem, for task distribution $p$ and horizon $H$.

**Theorem 4.1** *For* GMPS, *assuming reward-to-go bounded by $\delta$, and training error bounded by $\epsilon_{\theta*}$, we can show that* $\mathbb{E}_{i \sim p(\mathcal{T})}[\mathbb{E}_{\pi_{\theta + \nabla_\theta \mathbb{E}_{\pi_\theta}[R_i]}}[\sum_{t=1}^{H} r_i(\mathbf{s}_t, \mathbf{a}_t)]] \geq \mathbb{E}_{i \sim p(\mathcal{T})}[\mathbb{E}_{\pi_i^*}[\sum_{t=1}^{H} r_i(\mathbf{s}_t, \mathbf{a}_t)]] - \delta \sqrt{\epsilon_{\theta*}} O(H)$, *where $\pi_i^*$ are per-task expert policies.*

The proof of this theorem requires us to assume that the inner policy update in Eq. 2 can bring the learned policy to within a bounded error of each expert, which amounts to an assumption on the *universality* of gradient-based meta-learning [10]. The theorem amounts to saying that GMPS can achieve an expected reward that is within a bounded error of the optimal reward (i.e., the reward of the individual experts), and the error is linear in $H$ and $\sqrt{\epsilon_{\theta*}}$. The analysis holds for GMPS when each iteration generates samples by adapting the current meta-trained policy to each training task. However, we find in practice that the the initial iteration, where data is simply sampled from per-task experts $\pi_i^*$, is quite stable and effective; hence, we use this in our experimental evaluation. For the full proof of Theorem 4.1, see Appendix.

### 4.3 Algorithm Implementation

We next describe the full meta-RL algorithm in detail.

**Expert Policy Optimization.** The first phase of GMPS entails learning policies for each meta-training task. The simplest approach is to learn a separate policy for each task from scratch. This can already improve over standard meta-RL, since we can employ efficient off-policy reinforcement learning algorithms. We can improve the efficiency of this approach by employing a *contextual* policy to represent the experts, which simultaneously uses data from all of the tasks. We can express such a policy as $\pi_\theta(\mathbf{a}_t | \mathbf{s}_t, \omega)$, where $\omega$ represents the task context. Crucially, the context only needs to be known during *meta-training* – the end result of our algorithm, after the second phase, still uses raw task rewards without knowledge of the context at meta-test time. In our experiments, we employ this approach, together with soft-actor critic (SAC) [15], an efficient off-policy RL method.

For training the experts, we can also incorporate extra information during meta-training that is unavailable at meta-test time, such as knowledge of the state or better shaped rewards, when available. The former has been explored in single-task RL settings [23, 31], while the latter has been studied for on-policy meta-RL settings [14].

**Meta-Optimization Algorithm.** In order to *efficiently* optimize the meta-objective in Eq. 2, we adopt an approach similar to MAML. At each meta-iteration and for each task $\mathcal{T}_i$, we first draw samples $\mathcal{D}^{\text{tr}}_{\mathcal{T}_i}$ from the policy $\pi_\theta$, then compute the updated policy parameters $\phi_{\mathcal{T}_i}$ using the $\mathcal{D}^{\text{tr}}_{\mathcal{T}_i}$, then we update $\theta$ to optimize $\mathcal{L}_{\text{BC}}$, averaging over all tasks in the minibatch. This requires sampling from $\pi_\theta$, so for efficient learning, we should minimize the number of meta-iterations.

We note that we can take multiple gradient steps on the behavior cloning meta-objective in each meta-iteration, since this objective does not require on-policy samples. However, after the first gradient step on the meta-objective modifies the pre-update parameters $\theta$, we need to recompute the adapted parameters $\phi_i$ starting from $\theta$, and we would like to do so *without* collecting new data from $\pi_\theta$. To achieve this, we use an importance-weighted policy gradient, with importance weights $\frac{\pi_\theta(\mathbf{a}_t|\mathbf{s}_t)}{\pi_{\theta_{\text{init}}}(\mathbf{a}_t|\mathbf{s}_t)}$, where $\theta_{\text{init}}$ denotes the policy parameters at the start of the meta-iteration. At the start of a meta-iteration, we sample trajectories $\tau$ from the current policy with parameters denoted as $\theta = \theta_{\text{init}}$. Then, we take many off-policy gradient steps on $\theta$. Each off-policy gradient step involves recomputing the updated parameters $\phi_i$ using importance sampling:

$$\phi_i = \theta + \alpha \mathbb{E}_{\tau \sim \pi_{\theta_{\text{init}}}} \left[ \frac{\pi_\theta(\tau)}{\pi_{\theta_{\text{init}}}(\tau)} \nabla_\theta \log \pi_\theta(\tau) A_i(\tau) \right] \tag{3}$$

where $A_i$ is the estimated advantage function. Then, the off-policy gradient step is computed and applied using the updated parameters using the behavioral cloning objective defined previously: $\theta \leftarrow \theta - \beta \nabla_\theta \mathcal{L}_{\text{BC}}(\phi_i, \mathcal{D}^{\text{val}}_i)$. This optimization algorithm is summarized in Alg. 2.

## 5 Experimental Evaluation

We evaluate GMPS separately as a meta-reinforcement algorithm, and for learning fast RL procedures from multi-task demonstration data. We consider the following questions: As a meta-RL algorithm, (1) can GMPS meta-learn more efficiently than prior meta-RL methods? For learning from demonstrations, (2) does using imitation learning in the outer loop of optimization enable us to overcome challenges in exploration, and learn from sparse rewards?, and further (3) can we effectively meta-learn CNN policies that can quickly adapt to vision-based tasks?

To answer these questions, we consider multiple continuous control domains shown in Fig. 2.

### 5.1 Experimental Setup

**Sawyer Manipulation Tasks.** The tasks involving the 7-DoF Sawyer arm are performed with 3D position control of a parallel jaw gripper (four DoF total, including open/close). The Sawyer environments include:
- Pushing, full state: The tasks involve pushing a block with a fixed initial position to a target location sampled from a 20 cm × 10 cm region. The target location within this region is not observed and must be implicitly inferred through trial-and-error. The 'full state' observations include the 3D position of the end effector and of the block.
- Pushing, vision: Same as above, except the policy receives images instead of block positions.
- Door opening: The task distribution involves opening a door to a target angle sampled uniformly from 0 to 60 degrees. The target angle is not present in the observations, and must be implicitly

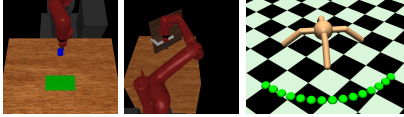

Figure 2: Illustration of pushing (left), door opening (center) and legged locomotion (right) used in our experiments, with the goal regions specified in green for pushing and locomotion.

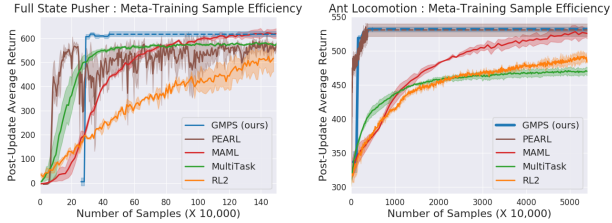

Figure 3: Meta-training efficiency on full state pushing and dense reward locomotion. All methods reach similar asymptotic performance, but GMPS requires significantly fewer samples.

inferred through trial-and-error. The 'full state' observations include the 3D end effector position of the arm, the state of the gripper, and the current position and angle of the door.

**Quadrupedal Legged Locomotion.** This environment uses the ant environment in OpenAI gym [3]. The ask distribution comprises goal positions sampled uniformly from the edge of a circle with radius 2 m, between 0 and 90 degrees. We consider dense rewards when evaluating GMPS as a meta-RL algorithm, and challenging sparse rewards setting when evaluating GMPS with demonstrations.

Further details such as the reward functions for all environments, network architectures, and hyperparameters swept over are in the appendix. Videos of our results are available online [1].

### 5.2 Meta-Reinforcement Learning

We first evaluate the sample efficiency of GMPS as a meta-RL algorithm, measuring performance as a function of the total number of samples used during meta-training. We compare to a recent inference based off-policy method (PEARL) [34] and the policy gradient version of model-agnostic meta-learning (MAML) [8], that uses REINFORCE in the inner loop and TRPO in the outer loop. We also compare to $RL^2$ [7], and to a single policy that is trained across all meta-training tasks (we refer to this comparison as MultiTask). At meta-training time (but not meta-test time), we assume access to the task context, i.e. information that completely specifies the task: the target location for the pushing and locomotion experiments. We train a policy conditioned on the target position with soft actor-critic (SAC) to obtain expert trajectories which are used by GMPS. The samples used to train this expert policy with SAC are included in our evaluation. At meta-test time, when adapting to new validation tasks, we *only* have access to the reward, which necessitates meta-learning without providing the task contexts to the policy.

From the meta-learning curves in Fig. 3, we see similar performance compared to PEARL, and 4x improvement for sawyer object pushing and about 12x improvement for legged locomotion over MAML in terms of the number of samples required. We also see that GMPS performs substantially better than PEARL when evaluated on test tasks which are not in the training distribution for legged locomotion

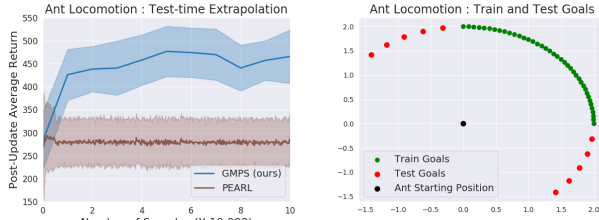

Figure 4: Test-time extrapolation for dense reward ant locomotion. The test tasks involve navigating to the red goals indicated (right). GMPS gets better average return across tasks (left).

(Fig. 4). This is because PEARL cannot generate useful contexts for out of distribution tasks, while GMPS uses policy gradient to adapt, which enables it to continuously make progress.

Hence, the combination of (1) an off-policy RL algorithm such as SAC for obtaining per-task experts, and (2) the ability to take multiple off-policy supervised gradient steps w.r.t. the experts in the outer loop, enables us to obtain significant overall sample efficiency gains as compared to on-policy meta-RL algorithm such as MAML, while also showing much better extrapolation than data-efficient contextual methods like PEARL. These sample efficiency gains are important since they bring us significantly closer to having a robust meta-reinforcement learning algorithm which can be run on physical robots with practical time scales and sample complexity.

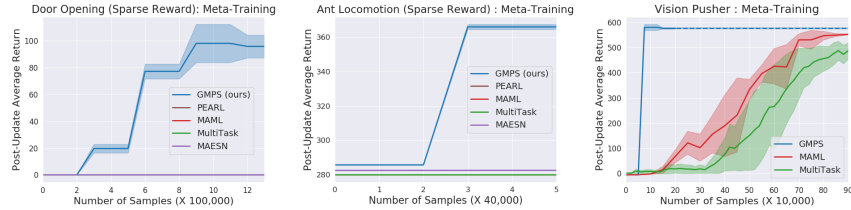

Figure 5: Meta-training comparisons for sparse reward door opening (left), sparse reward ant locomotion (middle) and vision pusher (right). Our method is able to learn when only sparse rewards are available for adaptation, whereas prior methods struggle. For vision-based tasks, we find that GMPS is able to effectively leverage the demonstrations to quickly and stably learn to adapt.

## 5.3 Meta-Learning from Demonstrations

For challenging tasks involving sparse rewards and image observations, access to demonstrations can greatly help with learning reinforcement learners. GMPS allows us to incorporate supervision from demonstrations much more easily than prior methods. Here, we compare against PEARL, MAML and MultiTask as in the previous section. When evaluating on tasks requiring exploration, such as sparse-reward tasks, we additionally compare against model agnostic exploration with structured noise (MAESN) [14], which is designed with sparse reward tasks in mind. Finally, we compare to a single policy trained with imitation learning across all meta-training tasks using the provided demonstrations (we refer to this comparison as MultiTask Imitation), for adaptation to new validation tasks via fine-tuning. For all experiments, the position of the goal location is not provided as input: the meta-learning algorithm must discover a strategy for inferring the goal from the reward.

**Sparse Reward Tasks.** One of the potential benefits of learning to learn from demonstrations is that exploration challenges are substantially reduced for the meta-optimizer, since the demonstrations provide detailed guidance on how the task should be performed. We hypothesize that in typical meta-RL, lack of easily available reward signal in sparse reward tasks makes meta-optimization very challenging, while using demonstrations makes this optimization signifi-

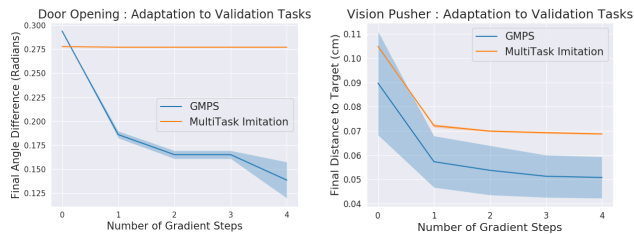

Figure 6: Comparison between GMPS and fine-tuning a policy pretrained with multi-task imitation, on held-out validation tasks for sparse-reward door opening (right) and vision pusher (left). By meta-learning the structure across tasks, GMPS achieves faster learning. Error bars are across different seeds.

cantly easier. To test this hypothesis, we experiment with learning to reinforcement learn from sparse reward signals in two different domains: door opening and sparse legged locomotion, as described in Section 5.1.

As seen from Fig. 5, unlike meta-RL methods such as MAESN, PEARL and MAML, we find that GMPS is able to successfully find a good solution in sparse reward settings and learn to explore. This benefit is largely due to the fact that we can tackle the exploration problem better with demonstrations than requiring meta-reinforcement learning from scratch. We also find that GMPS adapts to validation tasks more successfully than a policy pre-trained with MultiTask imitation (Fig. 6). The policy pre-trained with imitation learning does not effectively transfer to the new validation tasks via fine-tuning, since it is not trained for adaptability.

**Vision Based Tasks.** Deep RL methods have the potential to acquire policies that produce actions based simply on visual input [22, 25, 9]. However, vision based policies that can quickly adapt to new tasks using meta-reinforcement learning have proven to be challenging because of the difficulty of optimizing the meta-objective with extremely high variance policy gradient algorithms. On the other hand, visual imitation learning algorithms and RL algorithms that leverage supervised learning have been far more successful [23, 2, 13, 50], due to stability of supervised learning compared with RL. We evaluate GMPS with visual observations under the assumption that we have access to visual demonstrations for the meta-training tasks. Given these demonstrations, we directly train vision-based policies using GMPS with RL in the inner loop and imitation in the outer loop. To best leverage the added stability provided by imitation learning, we meta-optimize the entire policy (both fully connected and convolutional layers), but we only adapt the fully connected layers in the inner loop. This enables us to get the benefits of fast adaptation while retaining the stability of meta-imitation.

As seen in Fig. 5, learning vision based policies with GMPS is more stable and achieves higher reward than using meta-learning algorithms such as MAML. Additionally, we find that both GMPS and MAML are able to achieve better performance than a single policy trained with reinforcement learning across all the training tasks. In Fig. 6, we see that GMPS outperforms MultiTask Imitation for adaptation to validation tasks, just as in the sparse reward case.

## 6 Discussion and Future Work

In this work, we presented a meta-RL algorithm that learns efficient RL procedures via supervised imitation. This enables a substantially more efficient meta-training phase that incorporates expert-provided demonstrations to drastically accelerate the acquisition of reinforcement learning procedures and priors. We believe that our method addresses a major limitation in meta-reinforcement learning: although meta-reinforcement learning algorithms can effectively acquire adaptation procedures that can learn new tasks at meta-test time with just a few samples, they are extremely expensive in terms of sample count during meta-training, limiting their applicability to real-world problems. By accelerating meta-training via demonstrations, we can enable sample-efficient learning *both* at meta-training time and meta-test time. Given the efficiency and stability of supervised imitation, we expect our method to be readily applicable to domains with high-dimensional observations, such as images. Further, given the number of samples needed in our experiments, our approach is likely efficient enough to be practical to run on physical robotic systems. Investigating applications of our approach to real-world reinforcement learning is an exciting direction for future work.

## 7 Acknowledgements

The authors would like to thank Tianhe Yu for contributions on an early version of the paper. This work was supported by Intel, JP Morgan and a National Science Foundation Graduate Research Fellowship for Abhishek Gupta.

## Footnotes

[1] The website is at https://sites.google.com/berkeley.edu/guided-metapolicy-search/home

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
