[Supplementary Material]

# A  Theoretical Analysis

In this section, we provide a proof of Theorem 1, performing an analysis for the aggregation version of GMPS. However, note that our experiments find that the off-policy optimization with expert trajectories before any aggregation is also quite effective and stable empirically. First, we restate the theorem:

**Theorem 4.1** *For* GMPS, *assuming reward-to-go bounded by $\delta$, and training error bounded by $\epsilon_{\theta*}$, we can show that $\mathbb{E}_{i\sim p(\mathcal{T})}[\mathbb{E}_{\pi_{\theta+\nabla_\theta\mathbb{E}_{\pi_\theta}[R_i]}}[\sum_{t=1}^{H} r_i(\mathbf{s}_t,\mathbf{a}_t)]] \geq \mathbb{E}_{i\sim p(\mathcal{T})}[\mathbb{E}_{\pi_i^*}[\sum_{t=1}^{H} r_i(\mathbf{s}_t,\mathbf{a}_t)]] - \delta\sqrt{\epsilon_{\theta*}}O(H)$, where $\pi_i^*$ are per-task expert policies.*

We can perform a theoretical analysis of algorithm performance in a manner similar to [18]. Given a policy $\pi$, let us denote $d_\pi^t$ as the state distribution at time $t$ when executing policy $\pi$ from time 1 to $t-1$. We can define the cost function for a particular task $i$ as $c_i(\mathbf{s}_t,\mathbf{a}_t) = -r_i(\mathbf{s}_t,\mathbf{a}_t)$ as a function of state $\mathbf{s}_t$ and action $\mathbf{a}_t$, with $c_i(\mathbf{s}_t,\mathbf{a}_t) \in [0,1]$ without loss of generality. We will prove the bound using the notation of cost first, and subsequently express the same in terms of rewards.

Let us define $\pi_\theta + \nabla_i\pi_\theta = \pi_{\theta+\nabla_\theta\mathbb{E}_{\pi_\theta}[R_i]}$ as a shorthand for the policy which is obtained after the inner loop update of meta-learning for task $i$, with return $R_i$ during meta-optimization. This will be used throughout the proof to represent a one-step update on a task indexed by $i$, essentially corresponding to policy gradient in the inner loop. We define the performance of a policy $\pi_\theta(\mathbf{a}_t|\mathbf{s}_t)$ over time horizon $H$, for a particular task $i$ as:

$$J^i(\pi) = \sum_{t=1}^{H} \mathbb{E}_{\mathbf{s}_t\sim d_{\pi_\theta}^t}[\mathbb{E}_{\mathbf{a}_t\sim\pi_\theta(\mathbf{a}_t|\mathbf{s}_t)}[c_i(\mathbf{s}_t,\mathbf{a}_t)]].$$

This can be similarly extended to meta-updated policies as

$$J^i(\pi_\theta + \nabla_i\pi_\theta) = \sum_{t=1}^{H} \mathbb{E}_{\mathbf{s}_t\sim d_{\pi_\theta+\nabla_i\pi_\theta}^t}[\mathbb{E}_{\mathbf{a}_t\sim\pi_\theta+\nabla_i\pi_\theta}[c_i(\mathbf{s}_t,\mathbf{a}_t)]].$$

Let us define $J_t^i(\pi,\tilde{\pi})$ as the expected cost for task $i$ when executing $\pi$ for $t$ time steps, and then executing $\tilde{\pi}$ for the remaining $H-t$ time steps, and let us similarly define $Q_t^i(\mathbf{s},\pi,\tilde{\pi})$ as the cost of executing $\pi$ for one time step, and then executing $\tilde{\pi}$ for $t-1$ time steps.

We will assume the cost-to-go difference between the learned policy and the optimal policy for task $i$ is bounded: $Q_t^i(\mathbf{s},\pi_\theta,\pi^*) - Q_t^i(\mathbf{s},\pi^*,\pi^*) \leq \delta, \forall i$. This can be ensured by assuming universality of meta-learning [10].

When collecting data in order to perform the supervised learning in the outer loop of meta optimization, we can either directly use the 1-step updated policy $\pi_\theta + \nabla_i\pi_\theta$ for each task $i$, or we can use a mixture policy $\pi_j^i = \beta_j\pi_i^* + (1-\beta_j)(\pi_\theta + \nabla_i\pi_\theta)$, where $j$ denotes the current iteration of meta-training. This is very similar to the mixture policy suggested in the DAgger algorithm [36]. In fact, directly using the 1-step updated policy $\pi_\theta + \nabla_i\pi_\theta$ is equivalent to using the mixture policy with $\beta_j = 0, \forall j$. However, to simplify the derivation, we will assume that we always use $\pi_\theta + \nabla_i\pi_\theta$ to collect data, but we can generalize this result to full mixture policies, which would allow us to use more expert data initially and then transition to using on-policy data.

When optimizing the supervised learning objective in the outer loop of meta-optimization to obtain the meta-learned policy initialization $\pi_\theta$, we assume the supervised learning objective function error is bounded by a constant $D_{\text{KL}}(\pi_\theta + \nabla_i\pi_\theta||\pi_i^*) \leq \epsilon_{\theta*}$ for all tasks $i$ and all per-task expert policies $\pi_i^*$. This bound essentially corresponds to assuming that the meta-learner attains bounded training error, which follows from the universality property proven in [10].

Let $l_i(\mathbf{s},\pi_\theta + \nabla_i\pi_\theta,\pi_i^*)$ denote the expected 0-1 loss of $\pi_\theta + \nabla_i\pi_\theta$ with respect to $\pi_i^*$ in state $\mathbf{s}$: $\mathbb{E}_{\mathbf{a}_\theta\sim(\pi_\theta+\nabla_i\pi_\theta)(\mathbf{a}|\mathbf{s}),\mathbf{a}^*\sim\pi_i^*(\mathbf{a}|\mathbf{s})}[\mathbf{1}[\mathbf{a}_\theta \neq \mathbf{a}^*]]$. From prior work, we know that the total variation divergence is an upper bound on the 0-1 loss [27] and KL-divergence is an upper bound on the total variation divergence [33].

Therefore, the 0-1 loss can be upper bounded, for all $\mathbf{s}$ drawn from $\pi_\theta + \nabla_i \pi_\theta$:

$$l_i(\mathbf{s}, \pi_\theta + \nabla_i \pi_\theta, \pi_i^*) = \leq D_{\text{TV}}(\pi_\theta + \nabla_i \pi_\theta || \pi_i^*)$$
$$\leq \sqrt{D_{\text{KL}}(\pi_\theta + \nabla_i \pi_\theta || \pi_i^*)}$$
$$\leq \sqrt{\epsilon_{\theta *}}.$$

This allows us to bound the meta-learned policy performance using the following theorem:

**Theorem A.1** *Let the cost-to-go* $Q_t^i(\mathbf{s}, \pi_\theta + \nabla_i \pi_\theta, \pi_i^*) - Q_t^i(\mathbf{s}, \pi_i^*, \pi_i^*) \leq \delta$ *for all* $t \in \{1, ..., T\}, i \sim p(\mathcal{T})$. *Then in* GMPS, $J(\pi_\theta + \nabla_i \pi_\theta) \leq J(\pi_i^*) + \delta \sqrt{\epsilon_{\theta *}} O(H)$, *and by extension* $\mathbb{E}_{i \sim tasks}[J(\pi_\theta + \nabla_i \pi_\theta)] \leq \mathbb{E}_{i \sim tasks}[J(\pi_i^*)] + \delta \sqrt{\epsilon_{\theta *}} O(H)$

*Proof*:

$$J^i(\pi_\theta + \nabla_i \pi_\theta) = J^i(\pi_i^*) + \sum_{t=0}^{T-1} J_{t+1}^i(\pi_\theta + \nabla_i \pi_\theta, \pi_i^*) - J_t^i(\pi_\theta + \nabla_i \pi_\theta, \pi_i^*)$$

$$= J^i(\pi_i^*) + \sum_{t=1}^{H} \mathbb{E}_{\mathbf{s} \sim d_{\pi_\theta + \nabla_i \pi_\theta}^t}[Q_t^i(\mathbf{s}, \pi_\theta + \nabla_i \pi_\theta, \pi_i^*) - Q_t^i(\mathbf{s}, \pi_i^*, \pi_i^*)]$$

$$\leq J^i(\pi_i^*) + \delta \sum_{t=1}^{H} \mathbb{E}_{\mathbf{s} \sim d_{\pi_\theta + \nabla_i \pi_\theta}^t}[l_i(\mathbf{s}, \pi_\theta + \nabla_i \pi_\theta, \pi_i^*)] \qquad (4a)$$

$$\leq J^i(\pi_i^*) + \delta \sum_{t=1}^{H} \sqrt{\epsilon_{\theta *}} \qquad (4b)$$

$$= J^i(\pi_i^*) + \delta T \sqrt{\epsilon_{\theta *}}$$

Equation 4a follows from the fact that the expected 0-1 loss of $\pi_\theta + \nabla_i \pi_\theta$ with respect to $\pi_i^*$ is the probability that $\pi_\theta + \nabla_i \pi_\theta$ and $\pi_i^*$ pick different actions in $\mathbf{s}$; when they choose different actions, the cost-to-go increases by $\leq \delta$. Equation 4b follows from the upper bound on the 0-1 loss.

Now that we have the proof for a particular $i$, we can simply take expectation with respect to $i$ sampled from the distribution of tasks to get the full result.

*Proof*:

$$J^i(\pi_\theta + \nabla_i \pi_\theta) \leq J^i(\pi_i^*) + \delta T \sqrt{\epsilon_{\theta *}}$$
$$\implies E_{i \sim p(\text{tasks})}[J^i(\pi_\theta + \nabla_i \pi_\theta)] \leq E_{i \sim p(\text{tasks})}[J^i(\pi_i^*)] + \delta T \sqrt{\epsilon_{\theta *}} \qquad (5a)$$

Now in order to convert back to the version using rewards instead of costs, we can simply negate the bound, thereby giving us the original theorem 4.1, which states:

$$\mathbb{E}_{i \sim p(\mathcal{T})}[\mathbb{E}_{\pi_\theta + \nabla_\theta \mathbb{E}_{\pi_\theta}[R_i]}[\sum_{t=1}^{T} r_i(\mathbf{s}_t, \mathbf{a}_t)]] \geq \mathbb{E}_{i \sim p(\mathcal{T})}[\mathbb{E}_{\pi_i^*}[\sum_{t=1}^{H} r_i(\mathbf{s}_t, \mathbf{a}_t)]] - \delta \sqrt{\epsilon_{\theta *}} O(H)$$

.

# B    Reward Functions

Below are the reward functions used for each of our experiments.

- Sawyer Pushing (for both full state and vision observations)

$$R = -\|x_{obj} - x_{pusher}\|_2 + 100 \mid c - \|x_{goal} - x_{pusher}\|_2 \mid$$

where c is the initial distance between the object and the goal (a constant).

- Door Opening

$$R = \begin{cases} \mid 10x \mid & x \le x^* \\ \mid 10(x^* - (x - x^*)) \mid & x > x^* \end{cases}$$

where x is the current door angle, and $x^*$ is the target door angle

- Legged Locomotion (dense reward)

$$R = -||x - x^*||_1 + 4.0$$

where x is the location of centre of mass of the ant, $x^*$ is the goal location.

- Legged Locomotion (sparse reward)

$$R = \begin{cases} -||x - x^*||_1 + 4.0 & ||x - x^*||_2 \le 0.8 \\ -m + 4.0 & ||x - x^*||_2 > 0.8 \end{cases}$$

where x is the location of center of mass of the ant, $x^*$ is the goal location, and $m$ is the initial $\ell_1$ distance between $x$ and $x^*$ (a constant).

## C   Architectures

- State-based Experiments

Used a neural network with two hidden layers of 100 units with ReLU nonlinearities each for GMPS, MAML, multi-task learning, and MAESN. As shown in prior work [11], adding a bias transformation variable helps improve performance for MAML, so we ran experiments including this variation. [The bias transformation variable is simply a variable appended to the observation, before being passed into the policy. This variable is also adapted with gradient descent in the inner loop]. The learning rate for the fast adaptation step $(\alpha)$ is also meta-learned.

- Vision-based Experiments

The image is passed through a convolutional neural network, followed by a spatial soft-argmax [23], followed by a fully connected network block. The 3D end-effector position is appended to the result of the spatial soft-argmax, which is then passed through a fully connected neural network block. The convolution block is specified as follows: 16 filters of size 5 with stride 3, followed by 16 filters of size 3 with stride 3 , followed by 16 filters of size 3 with stride 1. The fully-connected block is as follows: 2 hidden layers of 100 units each. All hidden layers use ReLU nonlinearities.

## D   HyperParameters

The following are the hyper-parameter sweeps for each of the methods [run for each of the experimental domains] , run over 3 seeds.

1. GMPS
   (a) Number of trajectories sampled per task. : [20 , 50]
   (b) Number of tasks for meta-learning: [10 , 20]
   (c) Initial value for fast adaptation learning rate: [0.5, 0.1]
   (d) Variables included for fast adaptation: [all parameters, only bias transform variable]
   (e) Dimension of bias transform variable: [2, 4]
   (f) Number of imitation steps in between sampling new data from the pre-update policy: [1 , 200, 500, 1000, 2000]
2. MAML
   Hyper-parameter sweeps (a) - (d) from GMPS
3. MAESN
   Hyper-parameter sweeps (a) - (c) from GMPS

588         (a) Dimension of latent variable: [2,4]

589     4. MultiTask

590         (a) Batch size: [10000, 50000]

591         (b) Learning rate: [0.01, 0.02]

592     5. Contextual SAC [which is used to learn experts that are then used for GMPS]

593         (a) Reward scale: [10, 50, 100] (constant which scales the reward)

594         (b) Number of gradient steps taken for each batch of collected data: [1, 5, 10]