[Reviews · NeurIPS 2019]

Reviewer 1



The proposed method is a novel (and elegant) combination of existing techniques from off-policy learning, imitation learning, guided policy search, and meta-learning. The resulting algorithm is new and I believe it can be valuable to researchers in this field. I think the paper does not sufficiently discuss the related work of Rakelly et al [1] (PEARL). The paper is cited in the related work section as one of the methods based on "recurrent or recursive neural networks". If I remember correctly they don't use recurrency (only in one of their ablation studies to show that their method *outperforms* a recurrent-based encoder). Furthermore, this paper should be pointed out as being an *off-policy* meta-RL algorithm. While I think the methods are sufficiently different, this difference should be explained in the paper. I also believe that a comparison in terms of performance and sample efficiency (during meta-training) would be an interesting addition to the experimental evaluation. The quality and clarity of the paper high. The method is clearly motivated, and the technical contribution well explained. I think the pseudo-code helps a lot, and together with the supplementary material and code provided and expert reader should be able to reproduce the results. Overall, I think this paper has high significance. The proposed method is novel and well executed and there are multiple good ideas that go along with it. I'm surprised the paper does not compare to PEARL [1], but I believe this can be easily addressed. References: [1] Rakelly, Kate, et al. "Efficient off-policy meta-reinforcement learning via probabilistic context variables." arXiv preprint arXiv:1903.08254 (2019). ------------------------------------------------------------------------------------------------------ - UPDATE - I would like to thank the authors for their rebuttal and for adding the comparison to PEARL - these are very interesting insights! Together with these results, I think this is a strong submission and I therefore increase my score to 7, and vote for acceptance.

Reviewer 2



This work builds on the Model Agnostic Meta Learning (MAML) algorithm and adapts it to make it more sample efficient and able to use expert demonstrations. To the best of my knowledge, the work is original. The paper is well presented and polished. Notation is clear. The prose is easy to understand and I could see no obvious typos. The article is clear and well organized. The supplementary material is quite thorough at presenting information that could be used to reproduce the experimental results. I believe that this is an important work in the field of meta learning for reinforcement learning. Improving sample efficiency is key to make those algorithm useful in practice and this paper makes a clear contribution in that respect.

Reviewer 3



originality: This paper follows on from the state-of-the-art in meta-learning. Meta-learning consists in improving the sample efficiency of any reinforcement learning algorithms by finding an initial set of parameters from which it can quickly adapt to any task of interest. This paper proposes a new meta-learning algorithm that decouples meta-objective learning and meta-training by using two nested loops: the inner loop can be any reinforcement learning method, while the outer loop optimizes the meta-objective using a supervised learning method. It goes pretty much against the current trend of unified the deep learning black box and therefore is sufficiently original on its own. quality: The most significant application cases are discussed thoroughly and the authors provide strong theoretical convergence guarantees. However, the theoretical analysis covers only the particular case of on-policy learning, while the method is intended to be used for off-policy learning. The examples are appropriate to assess the performance of the method although they are quite simples and not very challenging control problems in the first place. I'm afraid it may still untractable in practice for real-world applications. clarity: The previous works are very well introduced. The relation with other works and the theoretical background is made very clear throughout the paper, making this paper readable for those without knowledge in meta-learning. The implementation details are very helpful and make the results pretty straight-forward to reproduce. significance: This paper is a valuable contribution to the literature on meta-learning, showing significant improvement over the state of the art. The idea of designing a meta-learning algorithm explicitly separating the meta-training and meta-objective learning into two distinct phases that can combine any reinforcement learning method to any supervised learning method is clearer and surprisingly effective. This architecture is very convenient to build on it, try different combinations, and improve its performance even further. It presents some theoretical convergence analysis, significant improvement of sample efficiency relative to some reference examples. It applies successfully to the challenging case of sparse reward, which is not the case of the other meta-learning methods, to my knowledge.

[Author Response · NeurIPS 2019]



Figure 1: Evaluation with added comparison to PEARL, showing meta-training curves on full state pushing (left), ant locomotion (middle), and sparse reward door opening (right). PEARL is more sample-efficient and achieves similar asymptotic performance on dense reward tasks. However, **GMPS significantly outperforms PEARL on sparse reward tasks.**

Figure 2: Test-time extrapolation for dense reward ant locomotion Left: Performance comparison. Right: Train and test goals. **GMPS is better able to learn out-of-distribution tasks.**

Figure 3: Ablation for number of consecutive outer updates, as requested by reviewer 3. Using 500 imitation steps (blue) results in significantly greater sample efficiency than using only one (pink).

We thank the reviewers for their positive and constructive feedback.

The primary concern from Reviewer 1 was the comparison to PEARL (Rakelly et al.). We have now added this
comparison. We performed this comparison for meta-training sample efficiency (Fig 1 left, middle), meta-training on
sparse reward tasks (Fig 1 right), and extrapolation to out of distribution tasks at test time (Fig 2).

With dense rewards, we observe PEARL and GMPS require about the same number of samples to meta-train, and
achieve similar performance (Fig 1 left, middle). On the sparse reward door task, we train PEARL in a setting that
matches ours: PEARL is trained with sparse rewards passed to the encoder during meta-training and meta-testing and
shaped rewards are used for meta-training the actor and critic weights. In this setting (Fig 1 right), PEARL is unable to
learn a strategy that explores sufficiently for the encoder to detect the sparse rewards. On out of distribution tasks (Fig 2
right), GMPS performs substantially better than PEARL (Fig 2 left). This is because GMPS uses policy gradient to
adapt, which enables it to continuously make progress on tasks even if they are out of distribution.

**Reviewer 1.** See PEARL comparisons above. We will also add a discussion of PEARL to the related work.

**Reviewer 2.** We will add a discussion of the algorithm's limitations and hyperparameter tuning to the revised paper. One
limitation for GMPS and for all current meta-RL methods is the difficulty in meta-training across qualitatively distinct
task families. This is due to two factors, the lack of benchmarks containing many different task families and because
learning only a few disjoint behaviors is challenging for a single neural network. The most important hyperparameters
to tune are the number of imitation steps per sampling step and the dimension of the bias transformation variable. We
will discuss alpha and beta in the revised version. Alpha is learned and beta is fixed at 0.01 across all experiments.

**Reviewer 3.** PEARL uses a different inner update rule than our algorithm (amortized inference instead of policy
gradient), and we show how this leads to worse extrapolation for PEARL (Fig. 2)

As requested, we added an ablation to show the effect of consecutive gradient steps between each outer iteration (Fig.
3), where we compare taking 500 imitation steps per sampling step (as in the paper) to taking only one imitation step
per sampling step (GMPS 1 outer). This results in poorer sample efficiency, since we no longer perform off-policy
gradient updates.

For ant locomotion, in the sparse setting, reward is provided only when the ant is within a certain distance of the goal.
Hence even if the ant performs the right behavior, its obtained return will be less than in the dense case since it receives
sparse reward for much of its trajectory.

[Meta-Review · NeurIPS 2019]

This paper has a valuable contribution to meta-learning and greatly improves the state-of-the-art. The proposed idea that explicitly separates the meta-learning algorithm and meta-objective learning into two distinct phases is significantly effective as shown in experiments. The idea is quite original compared to the current trend of unified the deep learning black box.Theoretical convergence analysis shows the proposed method significantly improved sample efficiency compared to some reference examples. The authors addressed a comparison to PEARL, adequately and the additional insight will strengthen the paper a lot. Given that the rebuttal was strong and the analysis well made, I found that the paper be accepted.